# Colorectal Cancer Detection Through Sweat Volatilome Using an Electronic Nose System and GC-MS Analysis

**DOI:** 10.3390/cancers17172742

**Published:** 2025-08-23

**Authors:** Cristhian Manuel Durán Acevedo, Jeniffer Katerine Carrillo Gómez, Gustavo Adolfo Bautista Gómez, José Luis Carrero Carrero, Rogelio Flores Ramírez

**Affiliations:** 1Multisensory Systems and Pattern Recognition Research Group, Faculty of Engineering and Architecture, University of Pamplona (UP), Pamplona 543050, Colombia; gustavo.bautista@unipamplona.edu.co (G.A.B.G.); jose.carrero@unipamplona.edu.co (J.L.C.C.); 2Chemical Engineering Group, Faculty of Engineering and Architecture, University of Pamplona (UP), Pamplona 543050, Colombia; 3Secretaria de Ciencia, Humanidades, Tecnología e Innovación, Research Fellow, Coordinación Para la Innovación Y Aplicación de la Ciencia Y la Tecnología (CIACYT), Avenida Sierra Leona No. 550, Colonia Lomas Segunda Sección, San Luis Potosí CP 78210, Mexico; rogelio.flores@uaslp.mx

**Keywords:** electronic nose, GC-MS, sweat, colorectal cancer, biomarkers, multivariate analysis, machine learning

## Abstract

Improving non-invasive, rapid, and accessible methods for colorectal cancer detection is critical for enhancing diagnostic accessibility and patient comfort. This study investigates the use of sweat analysis to identify volatile organic compounds (VOCs) associated with CRC and controls. 136 sweat samples were analyzed using gas chromatography–mass spectrometry (GC-MS) and a custom-designed electronic nose with 14 MEMS gas sensors. Multivariate analysis revealed clear discrimination between patients with CRC and healthy controls. Machine learning models achieved classification accuracies of up to 81% using GC-MS data and up to 97% using E-nose data. These findings underscore the potential of sweat-based VOC analysis as a reliable and non-invasive approach for CRC detection, with strong prospects for integration into early diagnostic workflows.

## 1. Introduction

Colorectal cancer (CRC) represents one of the most pressing global public health challenges. According to the Global Cancer Observatory (GLOBOCAN), 1,926,425 new cases were reported in 2022, accounting for 9.6% of the global cancer burden, and resulting in 904,019 associated deaths. These statistics rank CRC as the third most commonly diagnosed cancer and the second leading cause of cancer-related mortality worldwide [1]. Although historically more prevalent in older adults, recent years have seen a concerning rise in CRC incidence among individuals under 50, raising alarm within the medical community. This trend is associated with multiple risk factors, including diets high in animal fats, low fiber intake, smoking, sedentary lifestyle, obesity, age, genetic predisposition, family history, and alterations in gut microbiota [2]. By 2040, the global incidence of CRC is expected to continue rising, driven by population aging, demographic growth, and the increasing adoption of unhealthy lifestyles, particularly in low- and middle-income countries [3]. While some high-income countries have seen reduced CRC mortality due to early detection campaigns, public health education, and effective treatment options, significant disparities persist globally. For instance, countries with higher Human Development Index (HDI) scores report up to four times more new cases of CRC than those with fewer resources, reflecting deep inequities in access to preventive, diagnostic, and therapeutic strategies [4].

Colonoscopy is currently regarded as the gold standard method for CRC screening, as it enables direct visualization of the intestinal mucosa and removes precancerous polyps via endoscopic polypectomy [5]. However, its invasive nature, the requirement for bowel preparation, use of sedation, and the risk of complications such as bleeding or perforation limit patient compliance, especially in population-wide screening programs. To overcome these barriers, non-invasive methods have been developed, including fecal occult blood tests (FOBT), fecal immunochemical tests (FIT), multitarget stool DNA assays (e.g., Cologuard^®^), and CT colonography. Although these alternatives offer greater convenience and ease of implementation, they still suffer from limitations in sensitivity and specificity, particularly for detecting advanced adenomas, and are often unavailable in resource-constrained settings [6,7,8]. These limitations have spurred interest in developing non-invasive diagnostic technologies capable of detecting CRC in its early stages while being deployable in low-resource clinical environments [9]. In this context, the analysis of VOCs has emerged as a promising strategy, as VOCs have long been used to detect and monitor various physiological and pathological conditions. VOCs are low-molecular-weight (<300 Da) compounds with low polarity and high vapor pressure, produced as by-products of cellular metabolism under both normal and disease states. These compounds can be released and detected in various biological matrices, including exhaled breath, urine, feces, and sweat [10,11,12].

While breath [13,14,15,16,17,18,19,20,21,22,23,24,25,26,27,28], urine [29,30,31,32], and fecal [14,33,34] VOCs have shown promise in CRC [35] detection, human sweat remains a largely unexplored but highly promising matrix. Sweat contains molecules derived from the bloodstream and metabolic activity of skin microbiota. Eccrine sweat is predominantly composed of water and contains electrolytes (e.g., sodium, potassium, chloride, magnesium, calcium), along with amino acids, fatty acids, nitrogenous metabolites, and lipids, all of which provide valuable insights into an individual’s metabolic and physiological state [36,37]. Once secreted to the skin surface, sweat components may be metabolized by skin bacteria into VOCs, contributing to body odor. This combination of endogenous and microbial metabolites makes sweat a rich and dynamic source of volatile biomarkers [38,39,40].

Passive sweat collection through spontaneous perspiration, using absorbent patches or polymeric materials such as polydimethylsiloxane (PDMS), allows for non-invasive VOC sampling without the need for external stimulation or complex procedures. This method is painless, requires no preparation, enables real-time monitoring, and has demonstrated clinical applicability and reproducibility, providing practical advantages over other biological matrices [41,42].

Studies have focused on two main strategies for analyzing VOCs: instrumental techniques, such as GC-MS, and sensor-based systems, including E-noses. GC-MS remains the leading standard due to its high sensitivity, specificity, and ability to identify and quantify individual compounds in complex mixtures. However, it requires specialized equipment, skilled personnel, and lengthy processing times, limiting its suitability for routine clinical use [11,43]. In contrast, E-noses comprise arrays of chemical or physical sensors that detect VOC mixtures and generate electrical response patterns, which are then processed by artificial intelligence algorithms capable of recognizing disease-specific olfactory profiles [44]. Unlike GC-MS, E-noses do not identify individual compounds but capture global odor profiles, making them faster, portable, and scalable for large populations. In the context of CRC, E-noses have shown promising results in analyzing breath [45,46,47,48], urine [29,49,50], and feces [51,52], though their application to sweat remains unexplored [53].

Therefore, this study aims to advance the understanding of the sweat volatilome, a field that remains in its early stages of development, but also to explore a novel pathway for developing non-invasive, accessible, and potentially portable diagnostic methods to support early CRC detection. To this end, this research investigates for the first time the diagnostic potential of the human sweat volatilome for CRC detection using both GC-MS and an E-nose system based on MEMS technology. This investigation yielded four key findings:The E-nose achieved a classification accuracy of up to 97.1%, outperforming GC-MS in sensitivity, specificity, and overall predictive performance.GC-MS analysis identified 13 statistically significant volatile compounds that distinguished CRC patients from controls, indicating metabolomic alterations in sweat.Sweat, a biological matrix rarely explored in CRC diagnostics, contained relevant volatile biomarkers, underscoring its potential for non-invasive screening.The integration of multivariate analysis with machine learning algorithms enabled robust group discrimination and predictive modeling.

These findings position sweat volatilome analysis as a promising and practical tool for early, non-invasive detection of colorectal cancer.

## 2. Materials and Methods

Figure 1 illustrates the overall workflow schematic for analyzing sweat samples to detect CCR using electronic nose and GC-MS approaches.

### 2.1. Patient Selection

A total of 68 participants (aged 40–80 years) were recruited from Erasmo Meoz Hospital in Cúcuta, Colombia. Each volunteer provided two independent sweat samples, resulting in 136 biological records for analysis. Among them, 33 patients had histopathologically confirmed CRC via colonoscopy and biopsy, while the remaining 35 individuals formed the control group (i.e., volunteers without neoplastic pathology).

To ensure population homogeneity, exclusion criteria were applied. Patients who had undergone cancer treatment (radiotherapy or hormonal therapy), prostatectomy, or had urinary catheters or cancer in other organs were excluded. The study received approval from the hospital ethics committee and adhered to the ethical guidelines for human research. All participants gave written informed consent and were assigned anonymized ID codes to ensure confidentiality. Clinical questionnaires were used to collect medical data, including diagnosis. Additional information on comorbidities (e.g., diabetes, hypertension), medications, and lifestyle habits (e.g., smoking, alcohol) was recorded using a technical datasheet. Detailed volunteer characteristics of both the CRC and control groups are provided in Table 1.

### 2.2. Sample Collection

The day before the test, each patient received individualized instructions to ensure proper collection of sweat samples. Participants were advised to fast for at least 8 h before sampling and to refrain from consuming tobacco, alcohol, and certain foods the night before, as these factors could alter the volatile compound profiles. Additionally, they were instructed not to use perfumes, lotions, body creams, or aerosol deodorants to preserve the original chemical composition of the samples. Patients were also required to shower with neutral soap two hours before sample collection to minimize external contamination.

#### Sweat Sample Collection

Sweat sample collection was conducted using a standardized, at-home protocol designed to ensure consistency and minimize contamination. Each participant received a collection kit containing a sterile 5 cm × 5 cm gauze pad, a small sterile water dispenser, hypoallergenic micropore tape (3M), a pair of sterile nitrile gloves, and a labeled 15 mL Falcon tube. The night before sampling, participants cleansed the lower back (lumbar region) with sterile water only, avoiding soap or other chemical agents to preserve the skin’s natural chemistry. Once the area was completely dry, two sterile gauze pads were simultaneously affixed to the skin using micropore tape and left in place for approximately 8 h to collect spontaneous perspiration during sleep. The following morning, participants, wearing the provided gloves, carefully removed both gauze pads, placing each one into a separate, designated Falcon tube, and personally delivered the samples to the hospital laboratory for processing. The study was conducted in Cúcuta, Colombia, where nighttime temperatures commonly reach 27 °C or even higher. This ambient heat facilitated consistent perspiration across participants without requiring physical activity or artificial stimulation. In total, 136 sweat samples were collected from 68 participants; 68 samples were designated for GC-MS analysis, and the remaining 68 were used in the E-nose protocol. Upon reception, all samples were stored at −40 °C to preserve volatile content until further processing and analysis. It should be noted that all measurements were collected over two months.

Figure 2 depicts a symbolic image intended to illustrate the sweat collection procedure. It has been designed exclusively for illustrative purposes to support the methodology description and does not correspond to a real patient or clinical procedure.

### 2.3. GC-MS Chemical Analysis

As previously indicated, all sweat samples were frozen at −40 °C for approximately three months before undergoing instrumental analysis.

Before processing, the frozen samples were carefully thawed under controlled conditions (temperature and humidity) to preserve the chemical integrity of the VOCs. This thawing procedure involved maintaining the samples at 4 °C for approximately 12 h, followed by an additional thermal equilibration period at room temperature (15 to 20 min) to allow them to return to their original physiological state.

Once the conditioning process was complete, the samples were categorized and prepared for GC-MS analysis.

Sweat sample analysis was carried out using a gas chromatograph coupled with a mass spectrometer (GC–MS), Agilent 6890/5975C Series (Agilent Technologies, Santa Clara, CA, USA). Compound separation was achieved using a DB-5MS (Agilent Technologies, Santa Clara, CA, USA). capillary column (60.0 m × 250 μm × 0.25 μm), with helium as the carrier gas at a constant flow rate of 1.0 mL/min. The injector operated in splitless mode at 250 °C, while the mass spectrometer was set to SCAN mode, recording a mass range from 50 to 350 *m*/*z*. Thawed sweat samples were transferred from Falcon tubes into 15 mL cylindrical glass vials. Subsequently, 5 μL of a 0.1 ppm limonene solution was added to each sample as an internal standard. This compound was used to compensate for potential variations during injection, chromatographic separation, and GC-MS detection, as well as to facilitate system calibration and chromatographic peak correction. In this way, it was ensured that the compared signals corresponded consistently to the same compounds across all analytical runs and retention times. Finally, the vials were hermetically sealed with aluminum caps fitted with silicone septa until analysis.

Before analysis, thawed sweat samples were transferred from Falcon tubes into 15 mL cylindrical glass vials. Subsequently, 5 μL of a 0.1 ppm limonene solution was added to each sample as an internal standard. This compound was used to compensate for potential variations during injection, chromatographic separation, and GC-MS detection, as well as to facilitate system calibration and chromatographic peak correction. In this way, it was ensured that the compared signals corresponded consistently to the same compounds across all analytical runs and retention times. Finally, the vials were hermetically sealed with aluminum caps fitted with silicone septa until analysis. The vials were then heated in a sand bath at 60 °C for 5 min using a Hotplate (Corning PC-6200; Corning Inc., Corning, NY, USA) to promote the release of VOCs.

Analyte extraction was carried out using a SPME with a 50/30 μm divinylbenzene/carboxen/polydimethylsiloxane (DVB/CAR/PDMS) fiber mounted on a manual Stablex 24Ga (gray, 3pk) holder. The fiber was inserted through the vial septum and exposed for 12 min under the same thermal conditions to allow the adsorption of VOCs present in the sample’s headspace.

The fiber was then manually transferred to the GC injector for analysis, where it remained for 5 min to enable thermal desorption of the adsorbed compounds, initiating the chromatographic separation and detection process.

The oven temperature program was as follows: an initial temperature of 50 °C was maintained for 1 min, followed by a ramp of 5 °C/min to 150 °C, and then a further increase of 10 °C/min to a maximum of 230 °C. This temperature profile allowed the chromatographic run to be completed in 29 min, ensuring optimal separation of the volatile compounds present in the sweat matrix.

#### Volatilomic Data Analysis

Figure 3 illustrates two representative chromatographic signals, one from a CRC patient and the other from a control subject. Chromatographic data analysis was performed using MSD ChemStation software version E.02.02.1431, based on the spectra acquired through GC–MS. Peak alignment was carried out automatically using the software’s internal alignment tools, followed by manual verification to ensure accuracy, particularly for low-abundance peaks. The process then continued with the automatic identification of chromatographic peaks using the internal ‘ChemStation Integrator,’ which calculated the raw peak area (area under the curve, AUC) for each detected signal. These AUC values were used as the quantitative features for subsequent statistical and machine learning analyses after applying the selected normalization method. Preliminary compound identification was carried out by spectral comparison against the NIST98 and WILEY275 databases. The profiles illustrate the differences in volatile compound composition detected by GC–MS between the groups.

The results from each sample were exported as individual spreadsheets. Subsequently, retention times (RTs) were aligned across all samples, consolidating the data into a single table using retention time as the standard variable. This matrix was then subjected to a curation and dimensionality reduction process, which included:(a)The removal of retention times without relevant signal peaks.(b)Rounding of retention times to one decimal place to group redundant peaks and eliminate duplicates.

The final output was a curated and consolidated list of 68 samples and 187 variables corresponding to the selected retention times.

Subsequently, a custom script was developed in MATLAB R2024b (24.2 version), to process and analyze VOC biomarker data derived from the GC-MS system of sweat samples. The script was designed to automate the essential steps required for preparing metabolomics data for statistical modeling and classification. The workflow began with an interactive file selection interface, allowing the user to import a .csv file containing the complete data matrix, including sample identifiers, group labels (CRC vs. CO), and retention time variables.

Once imported, the script automatically parsed the group labels and removed non-numerical columns (e.g., identifiers), ensuring a clean feature matrix. Missing values were imputed using a moving average smoothing algorithm with a window size of three, enhancing the consistency of the dataset. A key module within the pipeline was the normalization interface, which provided a set of preprocessing techniques, including Min–Max scaling, Z-score standardization, quantile normalization, Pareto scaling, autoscaling, and logarithmic and square root transformations. These methods were implemented to minimize technical variability and ensure comparability across samples. In this study, autoscaling (mean-centered and scaled to unit variance) was selected based on its superior performance in subsequent multivariate classification tasks.

Following normalization, a feature selection step was implemented to identify the most relevant variables for distinguishing between groups, based on univariate statistical testing. The Wilcoxon rank-sum test was applied to each variable in the dataset to compare the distributions of the CRC and CO patients. This non-parametric test is particularly suited for detecting significant differences in small or non-normally distributed datasets.

Variables yielding a *p*-value < 0.05 were considered statistically significant and retained for further analysis. In cases where no features met this threshold, possibly due to overlapping distributions or limited sample size, an alternative mechanism was employed. Specifically, the algorithm automatically selected the top variables with the highest variance across all samples, assuming that greater variability may carry higher discriminatory potential.

This step output a reduced feature matrix containing only the most informative retention time features and their associated variable names.

### 2.4. Electronic Nose

This section describes the data acquisition process using the E-nose system for analyzing the sweat samples.

#### 2.4.1. Data Acquisition

For the E-nose measurements, 100 mL glass containers with silicone septa were used, into which the gauze containing the sweat sample was inserted. VOC extraction was performed from the headspace generated inside these containers by heating them on a hotplate DLAB MS7-H550-Pro at a controlled temperature of 40 °C for 10 min. Two sterile needles were placed to transfer the volatile compounds to the E-nose system: one allowed ambient air to enter the container, while the other directed the VOCs directly toward the electronic nose sensor.

It is important to note that an air filter (activated carbon) was placed at the inlet end of the ambient air line to eliminate gases, odors, and compounds that could alter the sweat sample. The acquisition protocol was divided into three main phases: (1) a sensor stabilization stage lasting 3 min, (2) a baseline acquisition for 1 min, and (3) a breath sample collection phase of 2 min. In addition, a 2 min sensor purge was conducted to remove VOC residues and restore baseline conditions. This purge was performed simultaneously with sensor stabilization, thereby not extending the total acquisition time.

Measurements were performed at 10:00 a.m. and concluded at approximately 12:00 p.m. Thus, each acquisition lasts approximately 6 min, and in total, all measurements were completed within an estimated cumulative time of 6 h and 30 min.

#### 2.4.2. E-Nose

Figure 4 shows the physical prototype and internal architecture of the E-nose system developed by the GISM research group from the Electronic Engineering Program at the University of Pamplona. This modular, multisensory device was specifically designed for headspace analysis of biological samples, such as sweat, in the context of non-invasive CRC detection. The system is portable, measuring 16 cm × 10 cm × 18 cm and weighing 1.6 kg.

Figure 4a shows the fully assembled system in operation, highlighting its compact white casing mounted on a stable base, a real-time graphical user interface for signal monitoring, and front-facing operational status indicators (cleaning, ready, measuring). The device integrates USB and sensor connectivity for seamless data acquisition and system control. A 1/4-inch transparent silicone tube is connected to the input port to transfer VOCs from sample containers into the system. On the other hand, Figure 4b provides a schematic representation of the internal components of the E-nose. The architecture consists of two primary chambers: (1) 100 mL sample conditioning chamber, which maintains VOCs under controlled conditions of temperature, humidity, pressure, and airflow; and (2) 10 mL sensor chamber housing 14 microelectromechanical system (MEMS)-based metal oxide (MOX) gas sensors selected for their high sensitivity and low power consumption.

The system is equipped with an electropump and an electrovalve that regulate the flow of VOCs from the sample container to the sensor chamber. A cooling fan is integrated into the system to prevent overheating of internal electronic components during prolonged operation, thus preserving sensor stability and accuracy. Additionally, an exhaust outlet allows the controlled release of VOCs after analysis, preventing cross-contamination between measurements. A dedicated power system, a controller, and a data acquisition board manage the operation and data transmission of the sensor array. This integrated configuration enables consistent VOC delivery, enhances thermal and analytical stability, and ensures reproducible measurements. Regarding the detection chamber, fabricated in stainless steel, it contains sensors distributed across two independent Printed Circuit Boards (PCBs), separating those with digital outputs from those with analog outputs. Each sensor is described with its model, associated label, number of units, manufacturer, and country of origin.

The digital sensors include BME680 (Bosch Sensortec, Reutlingen, Germany) with labels (1) BMU1 (BME680-U1) and (2) BMU5 (BME680-U5); CCS811 (ScioSense, Eindhoven, The Netherlands) with (3) CCU2 (CCS811-U2) and (4) CCU6 (CCS811-U6); and SPG30 (Sensirion, Stäfa, Switzerland) with (5) SGOH (SPG30-EtOH-U7), (6) SGH2 (SPG30-H2-U3), (7) SGO7 (SPG30-EtOH-U3), and (8) SGH7 (SPG30-H2-U7), respectively, deployed in duplicate. The analog sensors include MICS4514 (SGX Sensortech, Neuchâtel, Switzerland) with labels (9) VRED (VRED-U10) and (10) VOXU (VOX-U10); MICS6814 (SGX Sensortech, Neuchâtel, Switzerland) with (11) VNH3 (VNH3-U11) and (12) VNO2 (VNO2-U11); CCS801 (ScioSense, Eindhoven, The Netherlands) with (13) VOUT (VCO-U11); and GM502B (Winsen Electronics, Zhengzhou, China) with (14) VN12 (VNO2-U12), each used as a single unit. Thus, the sensor label corresponds to the specific VOC it is intended to detect, such as ethanol (EtOH), hydrogen (H_2_), ammonia (NH_3_), nitrogen dioxide (NO_2_), and other VOCs. This combination of sensor types and detection layers enables a broader VOC detection spectrum, thereby enhancing sensitivity and selectivity for distinguishing between CRC and CO samples.

To ensure repeatable results, all measurements were carried out in a controlled laboratory environment at 22 ± 1 °C and 45 ± 5% relative humidity. The conditioning chamber was designed to maintain standardized analysis conditions, providing a constant internal temperature of 40 °C and a relative humidity of approximately 30%.

Before each measurement, the samples underwent a preconditioning phase in which environmental variables (temperature and relative humidity) and system parameters (flow rate and pressure) were strictly controlled to ensure the repeatability and reliability of the results. During operation, the headspace sample was drawn from a glass container through a silicone tube connected to the device inlet. The VOCs were then directed into the conditioning chamber at a constant flow rate of 0.8 L/min, regulated by an electropump, and subsequently transferred to the sensor chamber. As mentioned, inside the conditioning chamber, the temperature was maintained at 40 °C and the relative humidity at 30% RH to ensure stable and reproducible measurement conditions. Sensor responses were recorded as the sample passed through the measurement chamber. For each acquisition, a 1 min baseline with ambient air was first established, followed by sample introduction, and then a 2 min cleaning phase with ambient air to purge residual VOCs and return the sensors to baseline.

Likewise, a similar data preprocessing strategy was applied to the E-nose system; however, it was implemented in Python 3.14 software to enable seamless hardware integration and real-time data acquisition. This setup also supports a graphical user interface for continuous monitoring of sensor signals. In contrast to GC-MS, the E-nose pipeline included an additional Orthogonal Signal Correction (OSC) step, which was crucial for reducing sensor drift, noise, and experimental artifacts inherent to gas sensor arrays. Using OSC enhanced the data’s robustness and interpretability, improving the separation between CRC and CO groups.

To mitigate the risk of overfitting, the OSC procedure was rigorously validated using cross-validation techniques. In addition to this correction step, the E-nose preprocessing pipeline mirrored the GC-MS workflow in terms of missing value handling, normalization options, and feature scaling procedures, ensuring methodological consistency across both platforms.

### 2.5. Data Analysis Methods

Multivariate analysis techniques were applied to both the GC-MS and E-nose datasets to explore the underlying data structure and enhance class separation. Specifically, Principal Component Analysis (PCA) and Partial Least Squares Discriminant Analysis (PLS-DA) were employed for the GC-MS data [54,55]. PCA served as an unsupervised method to reduce dimensionality and uncover intrinsic clustering patterns [56,57], while PLS-DA, a supervised technique [58,59], was employed to extract latent components that maximize the separation between CRC and CO groups.

In contrast, for the E-nose dataset, only PCA was applied due to its lower-dimensional structure, which facilitates exploratory visualization of sensor response patterns without class label supervision.

A range of machine learning models was employed for supervised classification to assess the discriminative power of the selected features. These included: Support Vector Machine (SVM) [60,61], which constructs optimal hyperplanes for binary classification; k-Nearest Neighbors (k-NN) [62,63], a distance-based non-parametric algorithm; Linear Discriminant Analysis (LDA); and Logistic Regression (LR), which incorporates penalty terms to prevent overfitting [64,65,66].

All models were trained and evaluated using stratified k-fold cross-validation to ensure robust and unbiased performance. Classification outcomes were evaluated using standard metrics, including accuracy, sensitivity, specificity, precision, and F1-score.

## 3. Results

### 3.1. GC-MS Dataset

As mentioned, a total of 187 compounds were detected in the sweat samples via GC–MS, from which some VOCs were identified as significantly different between patients with confirmed CRC and control subjects. Table 2 summarizes the results of the variable selection process, where each identified variable corresponds to an RT obtained from GC–MS analysis. For each RT, the MSD ChemStation software automatically integrated the chromatographic peak and calculated its AUC, which served as the quantitative value representing the relative abundance of that compound. In this table, the arrows indicate the direction of the abundance differences between the groups: an upward arrow shows a higher abundance in CRC compared to CO, while a downward arrow indicates a higher abundance in CO compared to CRC. These AUC values were then used as features for statistical and machine learning analyses. The selected features were ranked by their Variable Importance in Projection (VIP) scores, highlighting their ability to distinguish between CRC and CO groups. Higher VIP scores indicate greater relevance to the classification model. The table also includes the corresponding *p*-values, supporting the statistical significance of the observed differences.

To ensure data integrity and comparability across samples, the complete dataset was processed through a standardized preprocessing pipeline to enhance signal quality while minimizing technical noise and biological variability. In this case, the autoscaling method, where each variable is mean-centered and scaled to unit variance, produced the best discrimination between classes and was used throughout the analysis.

After normalization, statistically relevant features that differentiated between CRC and CO groups were identified using the Wilcoxon rank-sum test, as previously described. In cases where no variables met the significance threshold (*p* < 0.05), the thirteen compounds with the highest variance were retained to preserve discriminatory potential for downstream analysis. Figure 5 illustrates the top discriminant VOCs ranked by VIP scores for the two groups. The VIP scores derived from the PLS-DA model were used to differentiate between patients based on their VOC profiles. Each bar represents a single RT value corresponding to a detected VOC feature, with the bar length indicating the importance of that feature in the classification model.

Features with VIP scores greater than 0.1 are considered significant contributors to the group separation. The bars are color-coded according to the group in which the respective VOC showed higher mean abundance: red for CRC and blue for CO, thus providing insight into group-specific metabolic variations. For example, benzene, 1,3-dimethyl, tetradecanoic acid, 3,7-dimethylnonane, and undecane, 4,6-dimethyl, are the most influential VOCs associated with the control group, while styrene and octane, 2,4,6-trimethyl, are associated with CRC.

This analysis enables the identification of potential VOC biomarkers with high discriminatory power, which may help in the development of non-invasive diagnostic tools for CRC detection. The results highlight the metabolic alterations captured in the VOC profile that are unique to CRC patients compared to healthy controls.

After preprocessing, a PCA was performed to reduce dimensionality and uncover the latent structure within the dataset. Figure 6 shows the PCA plot derived from the sweat VOC profiles, which were analyzed using a custom-built script.

This plot illustrates the dimensionality reduction applied to the normalized dataset, visualizing the distribution of samples across the two groups.

It should be clarified that three biological sweat samples out of the initially acquired sixty-eight were excluded from the analysis due to experimental errors during the collection process. This decision was made to exclude potential outliers that could negatively impact the reliability and robustness of the data analysis; therefore, 65 measurements were used in this stage.

The resulting reduced feature matrix was projected into the principal component space, revealing clustering tendencies and underlying structure within the metabolomic data.

The PCA depicted differential metabolic profiles in sweat samples between patients with CRC and control subjects. As shown, the three principal components, PC1 (23.32%), PC2 (15.05%), and PC3 (12.11%), collectively account for approximately 50.48% of the total variance in the dataset. Although this cumulative variance was moderated, the plot reveals a noticeable visual separation between the CRC and CO clusters, indicating potential group discrimination.

The relatively low percentage of variance explained by the first three components indicates that the discriminatory information is spread across multiple dimensions, rather than being captured in just a few. This is a common characteristic of complex biological datasets, such as VOC metabolomic profiles, where subtle but biologically meaningful variations are distributed across multiple variables. Thus, while the variance captured here is not high, the PCA still provides valuable insight into the underlying group structure. The observed separation supports the hypothesis that sweat VOC profiles differ between groups.

While the overlap between groups implies shared metabolic signatures, the partial separation and density gradients observed between the groups support the hypothesis of condition-specific alterations in the sweat volatiles. This visual separation aligns with the PERMANOVA statistical result, which was used in this analysis, obtaining an F-value = 7.0, confirming that the differences between the two groups are statistically significant and not attributable to random variation.

These findings state that PCA, even in higher-order components, can reveal meaningful discrimination trends within complex metabolomic data, highlighting the diagnostic potential of sweat VOCs in colorectal cancer. On the other hand, the 3D PLS-DA plot depicted in Figure 7 demonstrates a clear supervised separation between the CRC and CO groups based on their VOC profiles.

Unlike PCA, an unsupervised method that identifies components capturing the most significant overall variance without regard to group membership, PLS-DA integrates class label information during the dimensionality reduction process.

This allows the algorithm to extract latent variables that are specifically optimized to maximize class separation.

The enhanced group discrimination observed in the PLS-DA space highlights the effectiveness of this supervised technique for modeling subtle yet biologically relevant differences in multivariate metabolomic data, making it a more suitable choice than PCA when the primary objective is classification or biomarker discovery.

In this visualization, the first three PLS components, PLS1 (35.37%), PLS2 (16.81%), and PLS3 (13.22%), account for a cumulative variance of 65.04%. Among them, PLS1 contributes the most to group separation, clearly distinguishing CRC from CO. Although PLS2 and PLS3 individually explain lower variance, they capture additional structure in the data that enhances the spatial separation of groups in three-dimensional space.

The distinct clustering pattern observed in this plot demonstrates the capability of PLS-DA to extract class-specific variance, even when the total explained variance is low. This highlights the strength of supervised multivariate models in capturing slight but biologically meaningful differences between the groups. The plot further supports the presence of discriminatory metabolic profiles in the sweat samples of CRC patients versus controls. After confirming the slight discrimination between the two groups, machine learning methods were applied to classify the measurements based on the information from the thirteen features.

The confusion matrix presented in Figure 8 illustrates the k-NN model’s classification performance in distinguishing CRC and CO samples based on three latent variables extracted through PLS-DA, using k = 5 as the number of nearest neighbors.

The evaluation was carried out using a 5-fold cross-validation strategy to ensure the robustness and generalizability of the model. A total of 65 sweat samples were analyzed, from which the model correctly classified 30 CO and 23 CRC samples. Misclassifications included 4 CO samples erroneously identified as CRC and 8 CRC samples misclassified as CO. These results translate into an overall classification accuracy of 81.5%. The specificity for the CO class was 88.2%, and the sensitivity for CRC was 74.2%. In terms of precision, 78.9% of the samples predicted as CO were truly CO, while 85.2% of those indicated as CRC were correctly identified. Figure 9 shows the ROC curve generated using the k-NN model for classifying both groups. The AUC value, approximately 0.81, indicates good discriminatory performance. AUC values close to 1 indicate high sensitivity and specificity and confirm the k-NN model’s robustness in distinguishing CRC from CO using the selected features.

Table 3 presents the classification accuracies obtained from four machine learning algorithms, using two types of input data: PCA scores and PLS-DA latent variables derived from GC-MS measurements. For PCA, the first three principal components (PC1–PC3) were calculated, with the corresponding scores used as model features. When these PCA scores were used as inputs, the performance across all models was moderate, with accuracies ranging from 70.8% (LDA) to 72.3% (LR, SVM, k-NN). In contrast, PLS-DA analyses were performed using up to three latent variables (PLS1–PLS3) as inputs, which significantly improved classification outcomes for all models. The highest accuracy was achieved by the k-NN classifier with 81.5%, followed by SVM (80.0%), LR (78.5%), and LDA (76.8%).

On the other hand, Table 4 summarizes the classification performance of the k-NN algorithm in distinguishing between the two groups of samples based on GC-MS data. The overall accuracy was comparable between both classes, with 81.5% for CO and 81.6% for CRC, indicating balanced performance.

The sensitivity was higher for CO (88.2%) than CRC (74.2%), signifying that the model was more effective at correctly identifying control samples. In contrast, the specificity was higher for CRC (88.2%) than CO (74.2%), indicating fewer false positives for CRC. Precision values were 78.9% for CO and 85.2% for CRC, showing a slightly stronger predictive confidence for CRC samples.

Thus, the F1-score, which balances precision and sensitivity, was 83.3% for CO and 79.3% for CRC, confirming good classification consistency across both groups. These results highlight the robustness of the k-NN model in handling the binary classification task, with slightly better sensitivity toward control samples and higher specificity and precision for CRC.

### 3.2. E-Nose Dataset

The results obtained using the E-nose to analyze sweat samples for classification are described below. Before PCA, as mentioned, the dataset was first processed to extract sensor features as the response amplitude (i.e., difference between maximum and minimum recorded signal, Max–Min) for each sensor, capturing the full dynamic range of the sensor response during sample exposure. The resulting dataset was then preprocessed using OSC to remove systematic variations unrelated to class separation, such as sensor drift and noise, and autoscaling to ensure equal weighting of all sensor variables.

Figure 10 displays two PCA score plots in 2D and 3D, illustrating the classification of CRC and CO samples based on E-nose data using 68 measurements. In plot (a), the first two principal components (PC1: 44.38%, PC2: 26.05%) account for 70.43% of the variance, facilitating partial separation between CRC and CO samples. Plot (b) incorporates a third component (PC3: 10.12%), increasing the cumulative explained variance to 80.55%. Including this third principal component improves the representation of the data structure. It reveals more distinct clustering patterns, reinforcing the discriminatory potential of E-nose signals for CRC detection.

Figure 11 presents the confusion matrix obtained from the classification of CRC and CO sweat samples using E-nose data analyzed with the LDA model. The model correctly classified 97.1% of CO samples and 97.0% of CRC samples, with only one sample misclassified in each group.

The normalized matrix highlights the model’s strong discriminative performance, with false positive and false negative rates of 2.9% and 3.0%, respectively. These values indicate a balanced and effective classification, demonstrating high sensitivity and specificity.

Achieving reliable separation between the groups, the classification model demonstrates excellent diagnostic performance, as reflected in its ROC curve (see Figure 12). The steep rise of the curve toward the top-left corner indicates strong sensitivity and specificity across a range of thresholds. In contrast, the AUC reaches 0.96, highlighting the model’s near-perfect discrimination capability. At its selected operating point, the model achieves a true positive rate of 0.97 and a false positive rate of only 0.03, indicating that it detects nearly all CRC cases while minimizing false positives.

Table 5 compares the classification performance of four machine learning models under two different data representations processed with OSC applied to raw normalized data and PCA-transformed scores. When using OSC-normalized data directly, classification accuracies ranged from 69.0% to 70.9%, with LDA performing slightly better than the other methods. However, when PCA was applied after OSC to project the data into a lower-dimensional space, a significant improvement was observed across all models. In this scenario, LDA achieved the highest accuracy at 97.1%, followed by k-NN (94.2%), LR (94.2%), and SVM (94.1%). These results demonstrate that combining OSC with PCA significantly enhances model performance, particularly for LDA, indicating that the transformed latent space more effectively captures the discriminative structure of the data for CRC detection.

The metrics were computed for both the CO and CRC groups to evaluate the model’s classification performance (see Table 6). The results demonstrate highly balanced and consistent performance across both categories.

The accuracy of 97.1% for both groups indicates that nearly all samples were correctly classified. The sensitivity was 97.0% for CRC, reflecting the model’s strong ability to correctly identify patients with the disease, while the CO group achieved a similarly high sensitivity of 97.1%. Specificity, which quantifies the model’s ability to correctly identify non-CRC cases, was 97.0% for controls and 97.1% for CRC, further supporting the model’s balanced classification capability. Additionally, precision and F1-score reached around 97.1% for CO and 97.0% for CRC, respectively, indicating that false positives were minimal and that the trade-off between accuracy and sensitivity was well-managed.

In this study, the E-nose demonstrated superior performance compared to GC-MS in differentiating CRC patients from control subjects based on sweat sample analysis. Although both techniques successfully identified relevant metabolic differences between the groups, the E-nose achieved higher sensitivity and overall classification accuracy.

## 4. Discussion

Sample acquisition and handling were key factors influencing the performance gap between the two platforms. The E-nose analyzed sweat samples directly and immediately, avoiding pre-treatment, storage, or transport, thus preserving VOC integrity. Even though sweat samples for the E-nose were stored briefly, the results indicate minimal degradation.

In contrast, GC-MS required several preparatory steps, including long-term storage, transport, and thermal desorption with SPME. These processes may have altered VOC composition through sample degradation or loss, especially the five-month storage.

It was observed that, in the initial sampling phase, the use of advanced analytical techniques, such as thermal desorption (TD) coupled with GC×GC/ToFMS, is crucial, as specific materials, including sterile gauze, can emit a wide range of volatile compounds. These emissions can interfere with the analytical process, complicating the identification and quantification of compounds. When compared with more specialized sampling phases, such as PowerSorb or passive sampling pillows (PSPs), gauze demonstrated lower efficiency in both VOC capture and release, as well as reduced reproducibility between replicates. Therefore, in CRC detection studies using GC-MS, adopting more efficient collection materials, such as PowerSorb, and incorporating preconcentration tools like Tenax tubes would be advisable. This would significantly enhance body volatilome (sweat) research reliability, particularly when high-resolution analytical techniques are employed [67]. Furthermore, the development and implementation of cleaner, thermally stable, and chemically versatile sampling phases are essential to ensure accurate and reproducible collection of volatile biomarkers [68].

Nevertheless, the results obtained in our pilot study support the feasibility of gauze-based sampling under controlled clinical workflows, particularly when coupled with conventional GC-MS or E-nose systems. Future efforts should focus on validating more thermally stable and cleaner sampling substrates to enhance reproducibility and expand clinical translation.

Additional variables such as SPME fiber type, desorption temperature, and chromatographic settings can further impact compound detection. While GC-MS confirmed metabolic differences between the groups, strict standardization of sample processing is crucial for consistent outcomes. On the other hand, the E-nose exhibited higher robustness and sensitivity in capturing real-time variations in VOC profiles.

Although sweat has been relatively underexplored as a diagnostic matrix, its easy accessibility, low cost, and non-invasive nature make it a promising alternative to more conventional biofluids such as breath, urine, or serum. Among the detected compounds, aliphatic and aromatic hydrocarbons were predominant, along with medium-chain fatty acids. Several of these compounds have been previously reported as potential biomarkers in various diseases and biological fluids, indicating the existence of a shared metabolomic profile across different pathological conditions. This overlap is of significant interest, as it suggests that the detected volatile compounds may reflect standard pathophysiological processes associated with tumor metabolism, such as oxidative stress, lipid peroxidation, and alterations in energy catabolism pathways [16,69,70,71,72,73,74].

Within the group of aromatic compounds, 1,3-dimethylbenzene stands out, as it has been associated with various types of cancer in previous studies, including lung, breast, prostate, colorectal, and leukemia. These compounds have primarily been detected in exhaled breath samples [26,75,76,77] and, in some cases, in feces, as potential markers of gastrointestinal diseases [78]. Their accumulation may be related to enzymatic dysregulation of hydrocarbon metabolism and environmental exposures that can enhance carcinogenic processes [79]. Similarly, styrene has been associated with liver toxicity, chronic respiratory diseases, renal cancer, type 2 diabetes, and prolonged exposure to cigarette smoke, highlighting its potential as a marker of generalized cellular damage [78,79,80,81,82].

Regarding branched and linear alkanes, compounds such as tetradecane, tridecane, 2-methyl; undecane, 4,6-dimethyl; pentadecane; n-heneicosane; n-pentacosane, and various isomers of octane (including 2,4,6-trimethyl, 4-methyl, and 5-ethyl-2-methyl) were detected. These compounds are recognized as byproducts of lipid peroxidation of cellular membranes and have been identified in various biological matrices from patients with different types of cancer. For example, n-heneicosane has been reported in the headspace of cultured pulmonary adenocarcinoma tumor cells, while tridecane, tetradecane, and octane, 4-methyl have been identified in patients with lung, gastric, and colorectal cancer, both in breath samples and tumor tissues. Additionally, some of these compounds have also been associated with chronic respiratory diseases such as asthma and COPD, as well as with liver cirrhosis [14,16,80,83,84,85,86,87,88].

Tetradecanoic acid (also known as myristic acid), a medium-chain saturated fatty acid, was also detected and has been reported in patients with prostate cancer [89], leukemia, lymphoma, carcinoma, type 2 diabetes [90], tuberculosis [91], colorectal cancer, both in fecal samples and in urine [35,92].

It is worth noting that some of the compounds identified in this study, such as octane, 2,4,6-trimethyl; tridecane, 2-methyl; and n-pentacosane, have not been previously associated directly with specific diseases according to the reviewed literature. However, their exclusive presence in the analyzed sweat samples indicates their potential role as emerging biomarkers related to CRC metabolism.

Additionally, statistical analysis using multivariate models identified several of these compounds as variables of importance for group discrimination, with VIP scores exceeding 0.4 in some cases. Specifically, benzene, 1,3-dimethyl showed a VIP score of 0.7798, followed by tetradecanoic acid (0.4595), 3,7-dimethylnonane (0.4070), and ethylbenzene (0.4026), reinforcing their potential role as discriminative markers between CRC patients and healthy controls.

## 5. Conclusions

This study highlights the potential of sweat volatilome analysis as a non-invasive approach for colorectal cancer detection, employing both GC-MS and an E-nose system. By profiling VOCs in sweat, both analytical platforms successfully discriminated between CRC patients and healthy controls, uncovering underlying metabolic alterations. The E-nose exhibited superior diagnostic performance, achieving higher classification accuracy, sensitivity, and specificity across multiple machine learning models compared to GC-MS. These results emphasize the advantages of the E-nose as a rapid, reliable, and operationally simple tool for early CRC screening.

In the case of GC-MS data, PLS-DA outperformed PCA, providing more apparent group separation and improved classification outcomes. While PCA provided an initial exploratory view of the data structure, the supervised nature of PLS-DA enabled it to capture class-specific variance better, thereby enhancing its discriminatory power. Meanwhile, the E-nose, when combined with data preprocessing techniques such as OSC and dimensionality reduction via PCA, achieved classification accuracies exceeding 97%, demonstrating balanced and robust performance across both CRC and control groups. In contrast, the GC-MS-based models yielded moderate performance, likely affected by VOC degradation and variability introduced during the more complex sample processing and storage steps.

These findings highlight the strong potential of the E-nose as a cost-effective, non-invasive diagnostic alternative that enables real-time analysis without requiring elaborate sample preparation, making it particularly suitable for clinical and point-of-care settings. Nevertheless, GC-MS remains valuable for in-depth biomarker discovery and chemical profiling, and future efforts to standardize its protocols could enhance its reproducibility and diagnostic utility in volatilomics-based cancer detection.

Future research should aim to expand the sample size, perform multi-center validation, and integrate complementary biosensing platforms to further validate and confirm the diagnostic potential of sweat-based volatilomics. In addition, the inclusion of high-risk adenomas in subsequent studies will be essential to assess screening applicability fully. Another limitation of this study was the absence of complete clinical information on CRC stage for all patients. As tumor stage may have influenced the volatilome profile, future studies should incorporate this variable to evaluate potential stage-related differences in diagnostic performance.

Ultimately, the strategic integration of a technological E-nose for rapid screening and GC-MS for detailed molecular profiling or biomarker detection could offer a comprehensive approach for early CRC detection and personalized health monitoring.

## Figures and Tables

**Figure 1 cancers-17-02742-f001:**
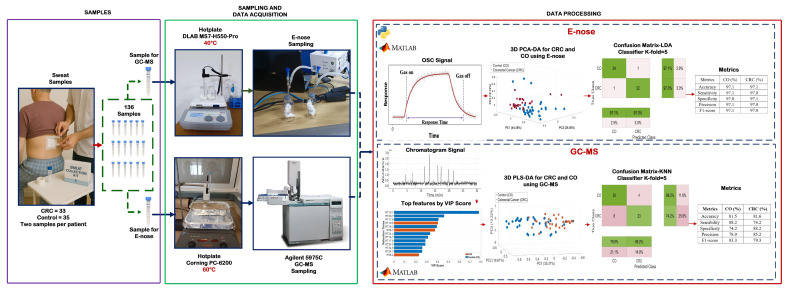
Scheme for CRC detection using sweat samples: sample collection, E-nose and GC-MS analysis, and data processing with multivariate analysis and machine learning.

**Figure 2 cancers-17-02742-f002:**
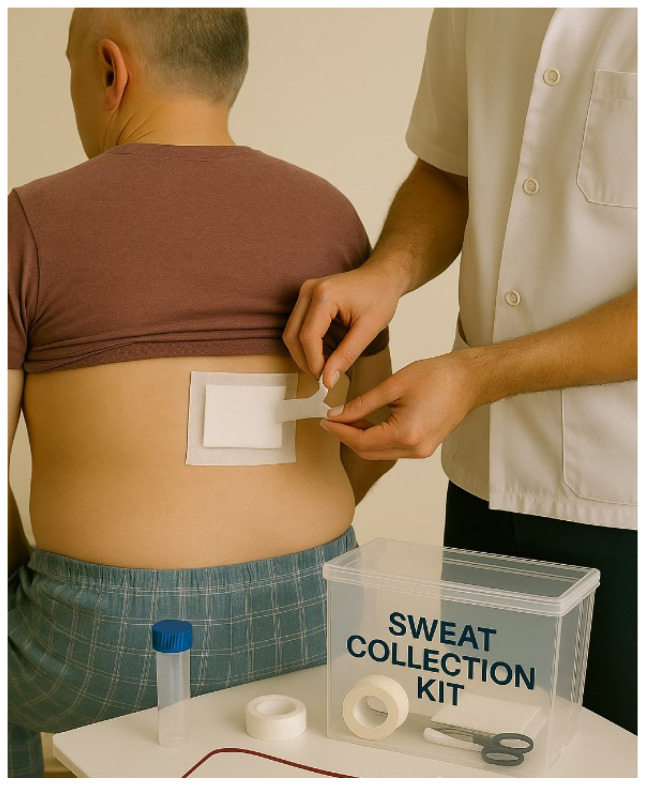
Sweat collection procedure for VOC analysis in CRC and CO samples.

**Figure 3 cancers-17-02742-f003:**
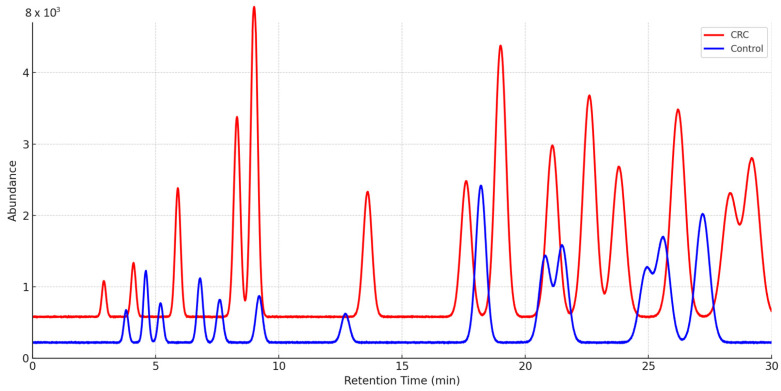
Representative chromatograms obtained from sweat samples of a CRC patient and a CO volunteer.

**Figure 4 cancers-17-02742-f004:**
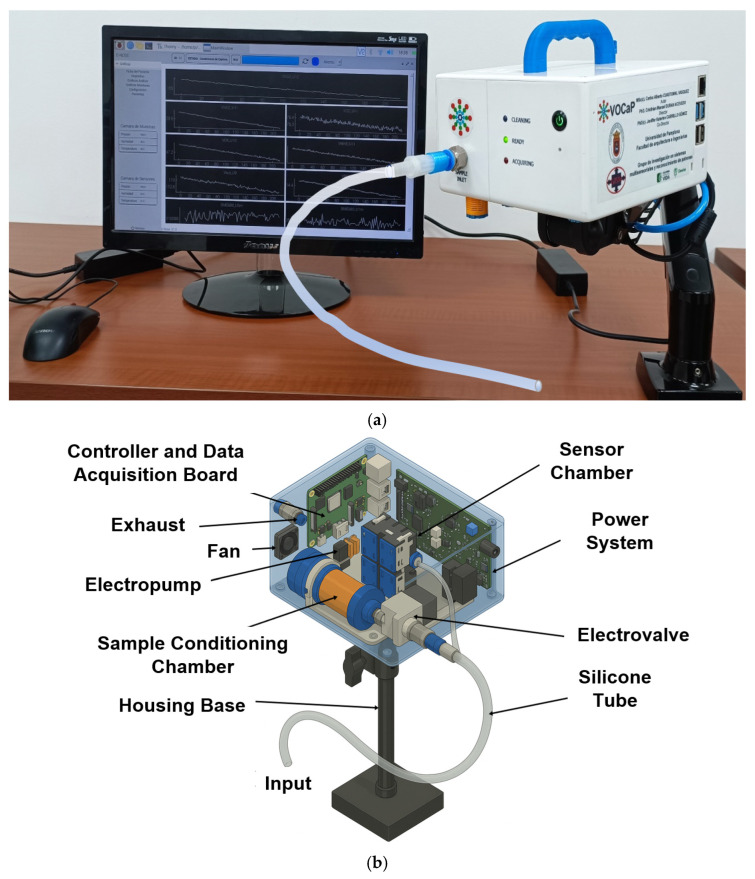
Portable E-nose for CRC and CO detection in sweat samples. (**a**) An external view of the E-nose system was developed for headspace VOC analysis. (**b**) An exploded schematic showing the main components, including the conditioning chamber, sensor array, electropump, electrovalve, and fan for cooling.

**Figure 5 cancers-17-02742-f005:**
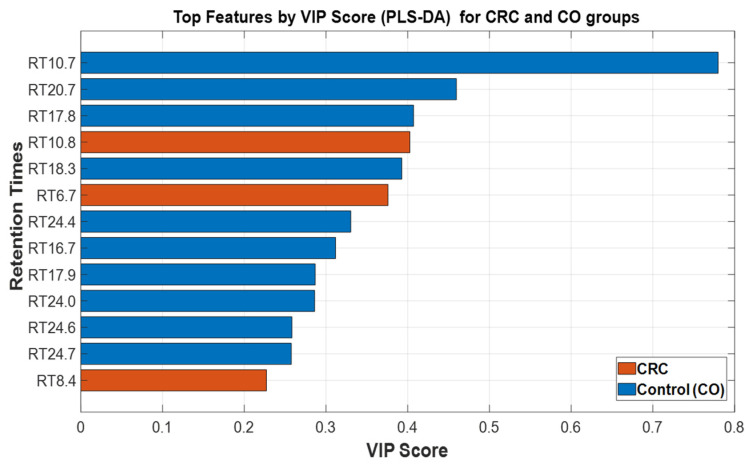
Top discriminant VOCs ranked by VIP Scores for CRC and CO Groups.

**Figure 6 cancers-17-02742-f006:**
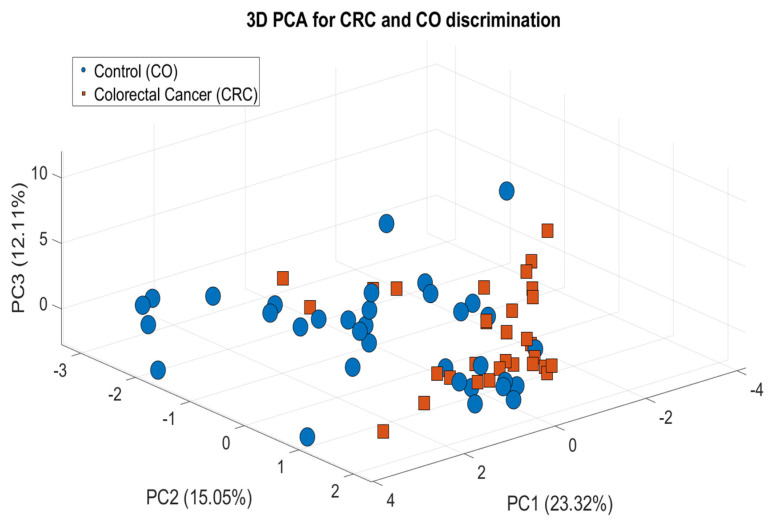
PCA-based visualization of GC-MS data for CRC vs. CO discrimination.

**Figure 7 cancers-17-02742-f007:**
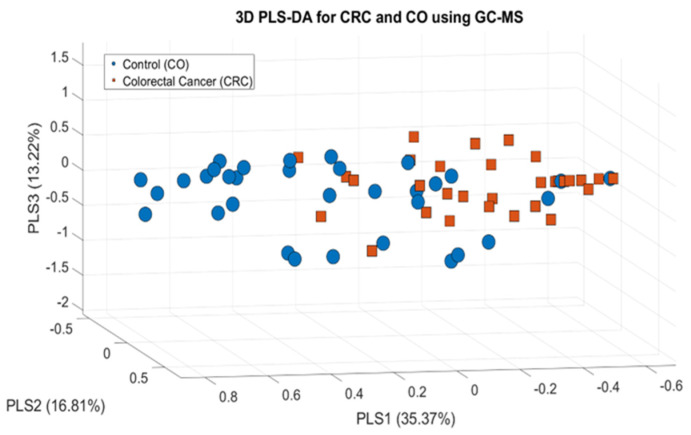
PLS-DA model for the discrimination of CRC and CO sweat samples based on GC-MS data.

**Figure 8 cancers-17-02742-f008:**
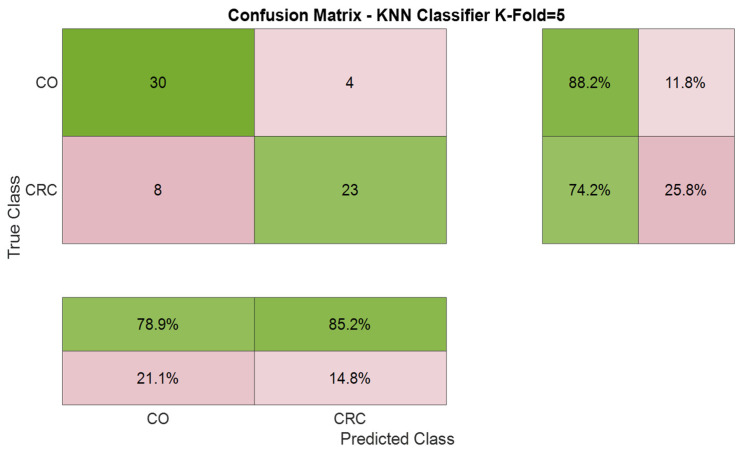
Confusion matrix obtained from the k-NN model to classify CRC and CO samples using GC-MS.

**Figure 9 cancers-17-02742-f009:**
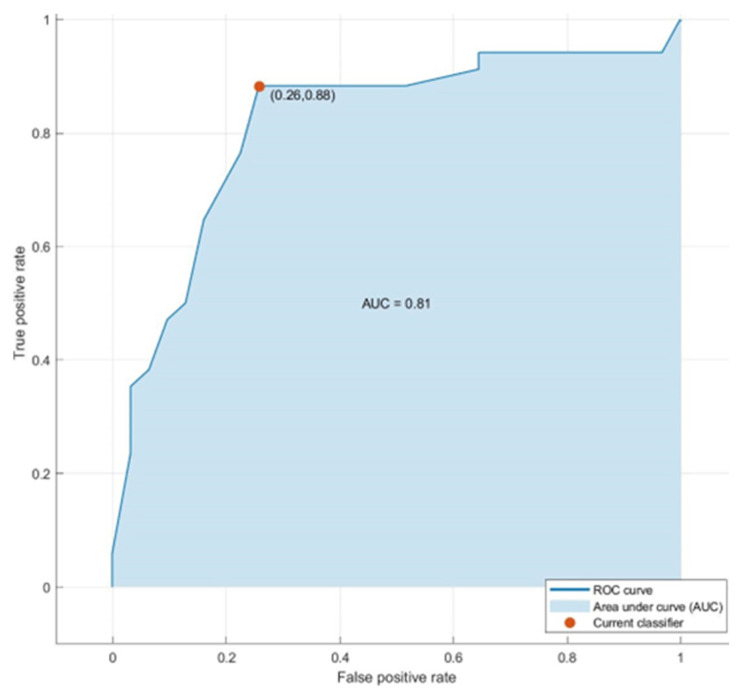
ROC curve and AUC for CRC detection using GC-MS data.

**Figure 10 cancers-17-02742-f010:**
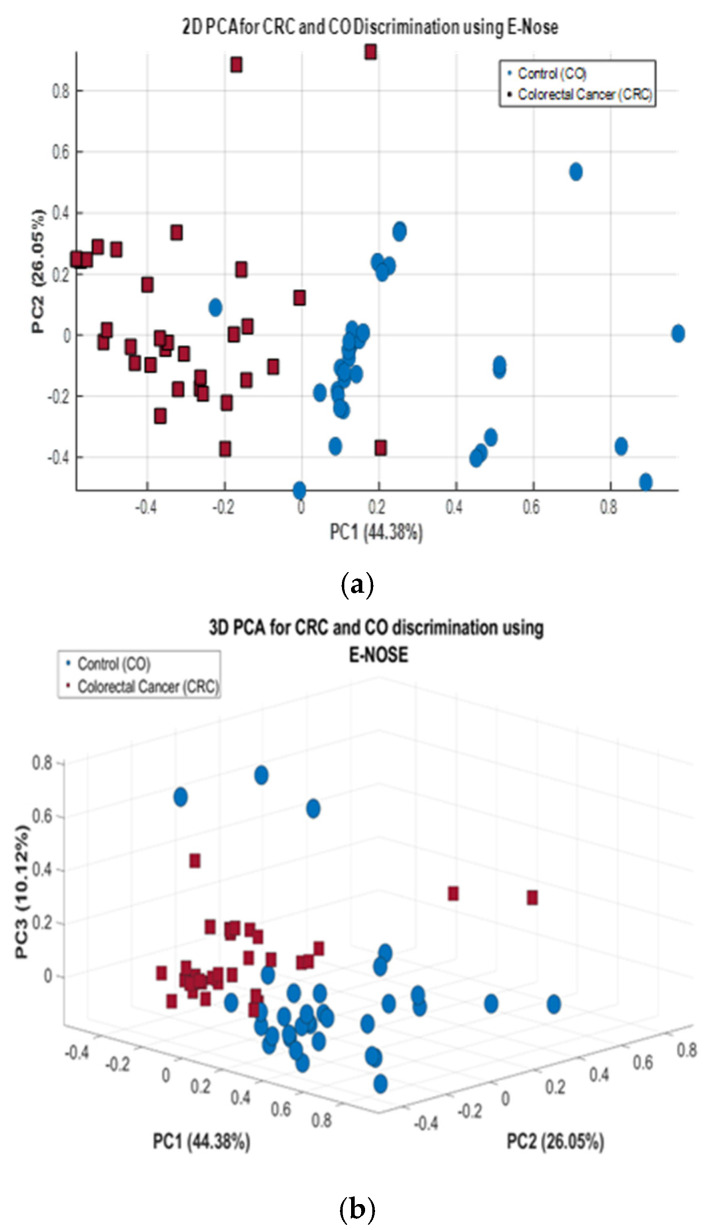
(**a**) 2D PCA plot for CRC and CO classification using E-nose data, (**b**) 3D PCA plot for CRC and CO classification using E-nose data.

**Figure 11 cancers-17-02742-f011:**
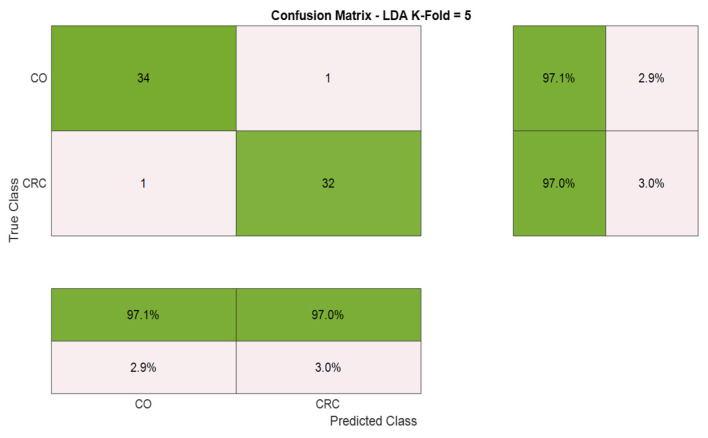
Confusion matrix obtained from the LDA model to classify CRC and CO samples using E-nose data.

**Figure 12 cancers-17-02742-f012:**
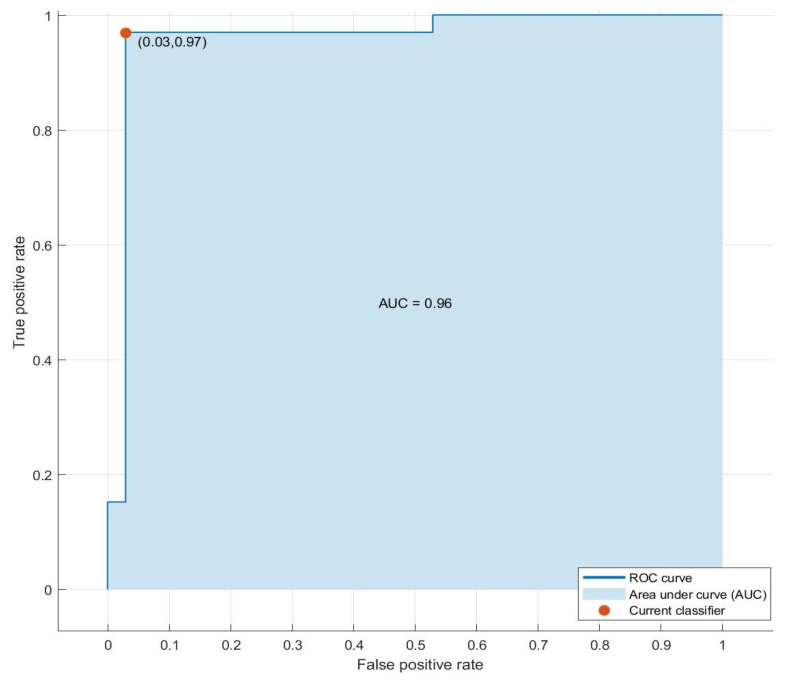
ROC curve and AUC values for CRC and CO detection obtained from LDA classification results using E-nose data from sweat samples.

**Table 1 cancers-17-02742-t001:** Characteristics of the volunteers in the CRC and control groups.

Volunteer ID	Gender		Age	Cluster		Smoker		Comorbidities	Number of Sweat Samples Collected
	F	M		CRC	CO	Yes	No		
CO001		x	64		x		x	Arterial hypertension	2
CO002	x		65		x		x	Hypothyroidism	2
CO003	x		76		x	x		Not detected	2
CO004		x	59		x		x	Arterial hypertension	2
CO005		x	58		x		x	Not detected	2
CO006	x		57		x		x	Not detected	2
CO007		x	51		x		x	Fatty liver, Hypercholesterolemia	2
CO008		x	65		x		x	Diabetes	2
CO009			53		x		x	Not detected	2
CO010		x	56		x		x	Not detected	2
CO011	x		68		x		x	Not detected	2
CO012			81		x		x	Diabetes, Arterial hypertension	2
CO013		x	52		x		x	Not detected	2
CO014	x		60		x		x	Arterial hypertension	2
CO015	x		59		x	x		Not detected	2
CO016		x	50		x		x	Gastritis, Hypothyroidism, Quadriplegia, Arterial hypertension	2
CO017		x	59		x		x	Diabetes	2
CO018	x		58		x		x	Not detected	2
CO019	x		51		x		x	Not detected	2
CO020		x	66		x		x	Arterial hypertension	2
CO021	x		52		x		x	Diabetes	2
CO022	x		51		x			Not detected	2
CO023	x		66		x		x	Not detected	2
CO024	x		52		x			Not detected	2
CO025	x		49		x			Not detected	2
CO026			40		x			Not detected	2
CO027	x		77		x		x	Not detected	2
CO028		x	45		x		x	Not detected	2
CO029		x	46		x		x	Not detected	2
CO030		x	51		x		x	Not detected	2
CO031		x	45		x	x		Not detected	2
CO032		x	56		x		x	Not detected	2
CO033	x		62		x		x	Not detected	2
CO034	x		58		x		x	Not detected	2
CO035		x	70		x		x	Not detected	2
CCR001		x	72	x			x	Not detected	2
CCR002	x		65	x			x	Not detected	2
CCR003	x		60	x			x	Arterial hypertension	2
CCR004	x		64	x			x	Arterial hypertension, Deep vein thrombosis	2
CCR005	x		60	x		x		Not detected	2
CCR006		x	76	x		x		Diabetes, Arterial hypertension	2
CCR007		x	67	x			x	Arterial hypertension	2
CCR008		x	48	x		x		Not detected	2
CCR009	x		48	x			x	Not detected	2
CCR010		x	70	x			x	Arterial hypertension	2
CCR011		x	65	x			x	Not detected	2
CCR012	x		49	x			x	Diabetic, Cholesterol	2
CCR013	x		55	x		x		Not detected	2
CCR014		x	40	x		x		Diabetes	2
CCR015		x	62	x			x	Arterial hypertension	2
CCR016		x	43	x			X	Not detected	2
CCR017		x	79	x			x	Arterial hypertension	2
CCR018		x	54	x		x		Arterial hypertension	2
CCR019	x		70	x			x	Not detected	2
CCR020	x		80	x			x	Not detected	2
CCR021		x	69	x			x	Not detected	2
CCR022		x	50	x		x		Not detected	2
CCR023		x	78	x		x		Not detected	2
CCR024	x		70	x			x	Arterial hypertension	2
CCR025	x		57	x			x	Not detected	2
CCR026		x	68	x		x		Not detected	2
CCR027		x	72	x		x		Arterial hypertension, Dyslipidemia, Diabetes	2
CCR028		x	58	x			x	Not detected	2
CCR029	x		47	x			x	Arterial hypertension	2
CCR030		x	78	x			x	Arterial hypertension	2
CCR031	x		49	x			x	Not detected	2
CCR032	x		43	x			x	Arterial hypertension	2
CCR033	x		59	x			x	Arterial hypertension, Diabetes	2

Volunteer ID, sex, age, diagnostic group (CRC: colorectal cancer; CO: control), smoking status, comorbidities, and number of collected sweat samples are shown. “x” indicates the presence of the respective condition or group.

**Table 2 cancers-17-02742-t002:** Selected GC-MS features with VIP scores and statistical differences between CRC and CO groups.

No	Retention Times (RTs)	ChemicalCompounds (Biomarkers)	VIPScore	AUC-CO	AUC-CRC	*p*-Value	Group
1	10.7	Benzene, 1,3-dimethyl	0.77980	1.87 × 10^6^	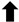	8.39 × 10^5^	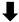	0.0096734	CO
2	20.7	Tetradecanoic acid	0.45952	1.83 × 10^6^	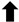	3.80 × 10^5^	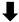	0.0114531	CO
3	17.8	3,7-Dimethylnonane	0.40707	9.22 × 10^5^	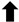	7.48 × 10^5^	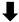	0.0286190	CO
4	10.8	Styrene	0.40255	5.58 × 10^6^	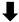	8.57 × 10^6^	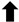	0.0288241	CRC
5	18.3	Undecane, 4,6-dimethyl	0.39271	4.55 × 10^5^	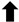	2.17 × 10^5^	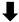	0.0068522	CO
6	6.7	Octane, 2,4,6-trimethyl	0.37585	2.90 × 10^8^	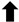	1.70 × 10^8^	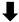	0.0333110	CRC
7	24.4	n-Heneicosane	0.33026	1.07 × 10^6^	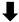	1.71 × 10^6^	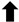	0.0086898	CO
8	16.7	Octane, 5-ethyl-2-methyl	0.31171	2.51 × 10^6^	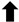	2.04 × 10^6^	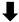	0.0211921	CO
9	17.9	Pentadecane	0.28676	3.65 × 10^5^	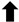	2.30 × 10^5^	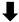	0.0416222	CO
10	24.0	Tridecane, 2-methyl	0.28066	2.65 × 10^5^	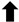	1.40 × 10^5^	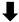	0.0296921	CO
11	24.6	Tetradecane	0.25827	1.34 × 10^5^	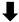	2.04 × 10^5^	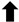	0.0183680	CO
12	24.7	n-Pentacosane	0.25752	4.22 × 10^5^	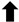	9.52 × 10^4^	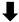	0.0142801	CO
13	8.4	Octane, 4-methyl	0.22711	5.27 × 10^5^	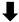	2.40 × 10^6^	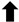	0.0369431	CRC

**Table 3 cancers-17-02742-t003:** Comparison of classification accuracies (%) obtained using four machine learning algorithms applied to PCA scores and PLS-DA latent variables derived from GC-MS data for CRC and CO sample discrimination.

Data	LDA (%)	LR (%)	SVM (%)	k-NN (%)
PCA (scores)	70.8	72.3	72.3	72.3
PLS_DA (Latent variables)	76.8	78.5	80.0	81.5

**Table 4 cancers-17-02742-t004:** Classification metrics for CRC and CO samples using the k-NN model based on GC-MS data.

Metrics	CO (%)	CRC (%)
Accuracy	81.5	81.6
Sensitivity	88.2	74.2
Specificity	74.2	88.2
Precision	78.9	85.2
F1-score	83.3	79.3

**Table 5 cancers-17-02742-t005:** Comparison of classification accuracies obtained using four machine learning algorithms, applied to data normalized with OSC and PCA scores derived from OSC-processed E-nose data, for the discrimination of CRC and CO samples.

Data	LDA (%)	LR (%)	SVM (%)	KNN (%)
Normalization (OSC)	70.9	69.1	69.0	69.1
PCA (scores) (OSC)	97.1	94.2	94.1	94.2

**Table 6 cancers-17-02742-t006:** Classification metrics for CRC and CO samples using the LDA model based on E-nose data.

Metrics	CO (%)	CRC (%)
Accuracy	97.1	97.1
Sensitivity	97.1	97.0
Specificity	97.0	97.1
Precision	97.1	97.0
F1-score	97.1	97.0

## Data Availability

The dataset is available upon request by emailing the authors.

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
