# Peer review of "Colorectal Cancer Detection Through Sweat Volatilome Using an Electronic Nose System and GC-MS Analysis"

_cancers, 2025, doi:10.3390/cancers17172742_

Round 1
Reviewer 1 Report
Comments and Suggestions for Authors
In the present manuscript, a study on the analysis of volatile organic compounds emitted from sweat for the diagnosis of colorectal cancer is reported. The analysis of volatile compounds was carried out using gas chromatography coupled with mass spectrometry and an array composed of 14 MEMS commercially available gas sensors. The topic is interesting, and the approach is innovative. Overall, the arguments are presented clearly and comprehensively. However, some aspects require attention. The main comments are as follows:
- Abstract: Line 41 – It would be appropriate to specify which machine learning technique, among those mentioned, was used to obtain the presented result.
- Section 2.2.1 – Sweat sample collection: The authors state that 136 samples were analyzed: 68 with GC-MS and 68 with the electronic nose. It would be appropriate to clarify how the sample duplicate was obtained: whether by splitting the gauze used or by applying two gauzes to the same subject simultaneously. Moreover, for clarity and scientific rigor, the body region where the sampling was performed should be specified, as this can influence sweat composition and therefore the volatilome.
- Section 2.3.1 – Volatilomic data analysis: It should be specified whether peak alignment was performed manually or using software.
- Section 2.4.1 – eNose: The authors aim to highlight the compactness of the device. If compactness is a key feature, please provide the device’s dimensions and weight.
It would also be appropriate to clarify the structure and functioning of the conditioning chamber during sample passage (p. 11, line 347) so that it is clearer how the sample is thermostated and conditioned inside.
Please also describe the sensor cleaning procedure used to establish their baseline and specify which carrier/reference gas was used. - Section 3.1 – GC-MS dataset: The authors are asked to specify which feature of the compounds was used for the analyses; it is assumed to be the AUC of the peak, but it would be better to explicitly state this in the text.
- Page 16, line 509: It is suggested that the authors report the number of PCA scores and PLS-DA latent variables used as input for the models.
- Lines 510 and 519: The results are reported as referring to eNose data, while the discussion is still about GC-MS data.
- Page 15, line 488: It is suggested to add the value of k (number of nearest neighbors) used in the k-NN analysis.
- Figure 8: Add labels to the values in the side tables to facilitate reading.
- Section 3.2 – eNose: Line 535 refers to exhaled breath and urine samples; this is presumably an error, since the entire work focuses on the sweat volatilome. Please also specify which type of sensor features were used for data processing.
Line 554 – Please clarify which type of data the presented results refer to.
In the caption of Figure 12, it is suggested to add that the ROC curve was obtained from the LDA results. - Discussion: Line 606 incorrectly reports that the analyzed sample was breath.
Author Response
Reviewer’s comments
We appreciated the editors' and reviewers' remarks, which helped us improve our manuscript. Based on their suggestions and comments, we have made several changes, using different colors to highlight the corrections.
Reviewer 1
In the present manuscript, a study on the analysis of volatile organic compounds emitted from sweat for the diagnosis of colorectal cancer is reported. The analysis of volatile compounds was carried out using gas chromatography coupled with mass spectrometry and an array composed of 14 MEMS commercially available gas sensors. The topic is interesting, and the approach is innovative. Overall, the arguments are presented clearly and comprehensively. However, some aspects require attention.
Response:
Thank you for the reviewer’s valuable comment and careful consideration of our work.
The main comments are as follows:
- Abstract: Line 41 – It would be appropriate to specify which machine learning technique, among those mentioned, was used to obtain the presented result.
Response: We sincerely appreciate this remark. In the revised abstract, we now specify all the machine learning techniques applied in this study and identify the classifiers that achieved the best performance for each data type. Specifically, the highest accuracy for GC-MS data was obtained using the k-NN classifier. In contrast, for E-nose data, the best results were achieved with the linear LDA classifier. This clarification enables readers to directly associate the reported results with the corresponding machine learning approach.
- Section 2.2.1 – Sweat sample collection: The authors state that 136 samples were analyzed: 68 with GC-MS and 68 with the electronic nose. It would be appropriate to clarify how the sample duplicate was obtained: whether by splitting the gauze used or by applying two gauzes to the same subject simultaneously. Moreover, for clarity and scientific rigor, the body region where the sampling was performed should be specified, as this can influence sweat composition and therefore the volatilome.
Response: We thank the reviewer for this precise observation.
The night before sampling, participants cleansed the lower back (lumbar region) with sterile water only, avoiding soap or other chemical agents to preserve the skin’s natural chemistry. Once the area was completely dry, two sterile gauze pads were simultaneously affixed to the skin using micropore tape and left in place for approximately 8 hours to collect spontaneous perspiration during sleep. The following morning, participants, wearing the provided gloves, carefully removed both gauze pads, placing each one into a separate, designated Falcon tube, and personally delivered the samples to the hospital laboratory for processing.
- Section 2.3.1 – Volatilomic data analysis: It should be specified whether peak alignment was performed manually or using software.
Response: We thank the reviewer for this relevant question
Chromatographic data analysis was performed using MSD ChemStation software based on the spectra acquired through GC–MS. Peak alignment was carried out automatically using the software’s internal alignment tools, followed by manual verification to ensure accuracy, particularly for low-abundance peaks.
- Section 2.4.1 – eNose: The authors aim to highlight the compactness of the device. If compactness is a key feature, please provide the device’s dimensions and weight.
It would also be appropriate to clarify the structure and functioning of the conditioning chamber during sample passage (p. 11, line 347) so that it is clearer how the sample is thermostated and conditioned inside.
Please also describe the sensor cleaning procedure used to establish their baseline and specify which carrier/reference gas was used.
Response: We thank the reviewer for this remark.
The eNose is a portable device measuring 16 cm × 10 cm × 18 cm and weighing 1.6 kg. During operation, the headspace sample was drawn from a glass container through a silicone tube connected to the device inlet. The VOCs were then directed into the conditioning chamber at a constant flow rate of 0.8 L/min, regulated by an electropump, and subsequently transferred to the sensor chamber. Inside the conditioning chamber, the temperature was maintained at 40 °C and the relative humidity at 30% RH to ensure stable and reproducible measurement conditions. Sensor responses were recorded as the sample passed through the measurement chamber. For each acquisition, a 1-minute baseline with ambient air was first established, followed by sample introduction, and then a 2-minute cleaning phase with ambient air to purge residual VOCs and return the sensors to baseline.
- Section 3.1 – GC-MS dataset: The authors are asked to specify which feature of the compounds was used for the analyses; it is assumed to be the AUC of the peak, but it would be better to explicitly state this in the text.
Response: We thank the reviewer for this remark.
Chromatographic data analysis was performed using MSD ChemStation software based on the spectra acquired through GC–MS. The process began with the automatic identification of chromatographic peaks using the internal ‘ChemStation Integrator,’ which calculated the raw peak area (area under the curve, AUC) for each detected signal. These AUC values were used as the quantitative features for subsequent statistical and machine learning analyses, after applying the selected normalization method.
Table 2 summarizes the results of the variable selection process, where each identified variable corresponds to an RT obtained from GC–MS analysis. For each RT, the MSD ChemStation software automatically integrated the chromatographic peak and calculated its AUC, which served as the quantitative value representing the abundance of that compound. These AUC values were then used as features for statistical and machine learning analyses.
- Page 16, line 509: It is suggested that the authors report the number of PCA scores and PLS-DA latent variables used as input for the models.
Response: We thank the reviewer for this important observation and support.
Table 3 presents the classification accuracies obtained from four machine learning algorithms, using two types of input data: PCA scores and PLS-DA latent variables derived from GC-MS measurements. For PCA, the first three principal components (PC1–PC3) were calculated, with the corresponding scores used as model features. When these PCA scores were used as inputs, the performance across all models was moderate, with accuracies ranging from 70.8% (LDA) to 72.3% (LR, SVM, k-NN). In contrast, PLS-DA analyses were performed using up to three latent variables (PLS1–PLS3) as inputs, which significantly improved classification outcomes for all models. The highest accuracy was achieved by the k-NN classifier with 81.5%, followed by SVM (80.0%), LR (78.5%), and LDA (76.8%).
- Lines 510 and 519: The results are reported as referring to eNose data, while the discussion is still about GC-MS data.
Response: We thank the reviewer for pointing out this inconsistency. In the original version, some results sections refer to eNose data incorrectly. We have carefully reviewed the text in lines 510 and 519 and corrected these errors so that the results and discussions are now consistently aligned with the corresponding dataset (GC–MS).
- Page 15, line 488: It is suggested to add the value of k (number of nearest neighbors) used in the k-NN analysis.
Response: We thank the reviewer for this important observation.
The confusion matrix presented in Figure 8 illustrates the k-NN model's classification performance in distinguishing CRC and CO samples based on three latent variables extracted through PLS-DA, using k = 5 as the number of nearest neighbors.
- Figure 8: Add labels to the values in the side tables to facilitate reading.
Response: We appreciate the reviewer’s suggestion to add labels to the side tables in Figure 8.
Unfortunately, MATLAB’s function does not currently provide a direct option to modify or rename the labels of the side summary tables (row- and column-normalized accuracies).
We apologize for any inconvenience this may cause.
- Section 3.2 – eNose: Line 535 refers to exhaled breath and urine samples; this is presumably an error, since the entire work focuses on the sweat volatilome.
Response: We thank the reviewer for this important observation.
The mention of exhaled breath and urine samples in Section 3.2, line 535, was indeed a mistake from an earlier draft. This study focuses exclusively on the sweat volatilome, and the text has been corrected accordingly to read sweat samples.
- Please also specify which type of sensor features were used for data processing.
Response: We thank the reviewer for this important observation.
Before PCA, as mentioned, the dataset was first processed to extract sensor features as the response amplitude (i.e., difference between maximum and minimum recorded signal, Max–Min) for each sensor, capturing the full dynamic range of the sensor response during sample exposure.
- Line 554 – Please clarify which type of data the presented results refer to.
Response: We thank the reviewer for this observation.
The results presented in line 554 refer specifically to eNose data obtained from sweat samples. The text has been revised accordingly to clarify the data source.
- In the caption of Figure 12, it is suggested to add that the ROC curve was obtained from the LDA results.
Response: We thank the reviewer for this observation.
We have revised the caption of Figure 12 to clearly indicate that the ROC curve and AUC values were obtained from LDA classification results based on eNose data. The updated caption now reads: Figure 12. ROC curve and AUC values for CRC and CO detection obtained from LDA classification results using eNose data from sweat samples.
- Discussion: Line 606 incorrectly reports that the analyzed sample was breath.
Response: We thank the reviewer for identifying this error. The text has been revised accordingly to replace ‘breath’ with ‘sweat’.
Reviewer 2 Report
Comments and Suggestions for Authors
The authors report a combined study involving GC-MS analysis and electronic nose measurements of sweat samples with the intention of non-invasive detection and analysis of colorectal cancer. 33 confirmed cases were compared to 35 controls. The work is of significance as it would indicate that there is change in volatile composition between the CRC-positive and negative samples which may be a very useful indicator. For use as a screening methods however, the authors should consider the presence of high-risk adenomas rather than just CRC positive patients. The stage of cancer for each patient is not reported. The GC data shown in Figure 3 are highly correlated - and it seems that the differences seen are in very small peaks. The GC-MS data reported should report the relative abundance. The sample sizes are small and there is no independent verification of the classifiers on previously unseen samples. It would have been good to separate the data into two separate training and test sets rather than the stratified approach used. As sample conditioning is highly important to attain repeatable results, it would be good to document in more detail how this was done - humidity, temperature, time of equilbration especially for the e-nose analysis. As there is always drift in sensors and sometimes problems in baseline recovery with e-nose systems - how were these systematic problems addressed. Over what time were the samples measured? Was a calibration standard used? Was there an internal standard used for the GC samples for peak alignment? How do the data in Table 2 compare with compounds detected in other studies? Table 4 - sensibility should be sensitivity - this is correct in Table 5. For references - there are a number of studies using urinary volatiles for CRC detection - these should also be quoted.
Author Response
Reviewer’s comments
We appreciated the editors' and reviewers' remarks, which helped us improve our manuscript. Based on their suggestions and comments, we have made several changes, using different colors to highlight the corrections.
Reviewer 2
The authors report a combined study involving GC-MS analysis and electronic nose measurements of sweat samples with the intention of non-invasive detection and analysis of colorectal cancer. 33 confirmed cases were compared to 35 controls. The work is of significance as it would indicate that there is change in volatile composition between the CRC-positive and negative samples which may be a very useful indicator.
Response: Thank you for the reviewer’s valuable comment and careful consideration of our work.
- For use as a screening methods however, the authors should consider the presence of high-risk adenomas rather than just CRC positive patients.
Response: We thank the reviewer for this valuable observation and fully agree. Our current cohort includes only CRC cases and colonoscopy and biopsy-negative controls; high-risk/advanced adenomas were not available. We have therefore acknowledged this limitation and included the inclusion of high-risk adenomas as a priority in future work, as stated in the conclusions section.
- The stage of cancer for each patient is not reported.
Response: We appreciate the reviewer’s comment. Information on cancer stage was not available for all CRC patients in this dataset, and staging was therefore not included in the current analysis. We recognize that tumor stage could influence VOC profiles and that its inclusion would add value to the interpretation of our results. We have now noted this as a limitation in the conclusions section and highlighted it as an essential variable to collect and analyze in future studies.
- The GC data shown in Figure 3 are highly correlated and it seems that the differences seen are in very small peaks.
Response: We thank the reviewer for this valuable observation.
Figure 3 has been replaced and improved to enhance the visualization of CRC and Control chromatograms.
- The GC-MS data reported should report the relative abundance.
Response: We appreciate the reviewer’s observation.
Table 2 summarizes the results of the variable selection process, where each identified variable corresponds to an RT obtained from GC–MS analysis. For each RT, the MSD ChemStation software automatically integrated the chromatographic peak and calculated its AUC, which served as the quantitative value representing the relative abundance of that compound. In this table, the arrows indicate the direction of the abundance differences between groups: an upward arrow shows a higher abundance in CRC compared to CO, while a downward arrow indicates a higher abundance in CO compared to CRC.
- The sample sizes are small, and there is no independent verification of the classifiers on previously unseen samples.
Response: We thank the reviewer for this observation. We acknowledge that the sample size in the present study is limited and that the classifiers were evaluated using internal cross-validation k-fold=5, without an independent external dataset.
As discussed in the Conclusions section, future work will focus on increasing the cohort size and performing multi-center studies, including validation with independent external samples, to further verify and strengthen the generalizability of the proposed models.
- It would have been good to separate the data into two separate training and test sets rather than the stratified approach used.
Response: We thank the reviewer for this observation. We acknowledge that the sample size in the present study is limited and that the classifiers were evaluated using internal cross-validation k-fold=5, without an independent dataset.
- As sample conditioning is highly important to attain repeatable results, it would be good to document in more detail how this was done: humidity, temperature, time of equilbration especially for the e-nose analysis, as there is always drift in sensors and sometimes problems in baseline recovery with e-nose systems. How were these systematic problems addressed.
Response: Thank you for pointing this out. In the revised manuscript, we have described the e-nose analysis procedure to include these details.
To ensure repeatable results, all measurements were carried out in a controlled laboratory environment at 22 ± 1 °C and 45 ± 5 % relative humidity. The conditioning chamber was designed to maintain standardized analysis conditions, providing a constant internal temperature of 40 °C and a relative humidity of approximately 30 %. Before each measurement, the samples underwent a preconditioning phase in which environmental variables (temperature and relative humidity) and system parameters (flow rate and pressure) were strictly controlled to ensure the repeatability and reliability of the results.
During operation, the headspace sample was drawn from a glass container through a silicone tube connected to the device inlet. The VOCs were then directed into the conditioning chamber at a constant flow rate of 0.8 L/min, regulated by an electropump, and subsequently transferred to the sensor chamber. As mentioned. inside the conditioning chamber, the temperature was maintained at 40 °C and the relative humidity at 30% RH to ensure stable and reproducible measurement conditions. Sensor responses were recorded as the sample passed through the measurement chamber. For each acquisition, a 1-minute baseline with ambient air was first established, followed by sample introduction, and then a 2-minute cleaning phase with ambient air to purge residual VOCs and return the sensors to baseline.
Likewise, a similar data preprocessing strategy was applied to the E-nose system; however, it was implemented in Python 3.14 software to enable seamless hardware integration and real-time data acquisition. This setup also supports a graphical user interface for continuous monitoring of sensor signals. In contrast to GC-MS, the E-nose pipeline included an additional Orthogonal Signal Correction (OSC) step, which was crucial for reducing sensor drift, noise, and experimental artifacts inherent to gas sensor arrays. Using OSC enhanced the data's robustness and interpretability, improving the separation between CRC and CO groups.
- Over what time were the samples measured?
Response: We thank the reviewer for the observation and would like to clarify the timeline. The complete acquisition of all measurements was carried out over approximately two months. On the other hand, all measurements were performed from 10:00 am to 12:00 pm.
Regarding GC-MS, in line 224, the reported run time of 29 minutes refers exclusively to the chromatographic acquisition time.
- Was a calibration standard used?
Response: We thank the reviewer for this remark.
Before analysis, thawed sweat samples were transferred from Falcon tubes into 15 mL cylindrical glass vials. Subsequently, 5 μL of a 0.1 ppm limonene solution was added to each sample as an internal standard. This compound was used to compensate for potential variations during injection, chromatographic separation, and GC-MS detection, as well as to facilitate system calibration and chromatographic peak correction. In this way, it was ensured that the compared signals corresponded consistently to the same compounds across all analytical runs and retention times. Finally, the vials were hermetically sealed with aluminum caps fitted with silicone septa until analysis.
- Was there an internal standard used for the GC samples for peak alignment?
Response: We thank the reviewer for this question. The corresponding explanation is provided in point 9), in a manner consistent with point 10).
- How do the data in Table 2 compare with compounds detected in other studies?
Response: We thank the reviewer for this observation. Although there is literature reporting these compounds in other pathologies such as lung, prostate, colorectal cancers, and leukemias, those studies have focused on different biological matrices, including breath, feces, and urine. This analysis is addressed in the Discussion section. Moreover, these findings highlight the potential relevance of these compounds as disease biomarkers and emphasize the importance of further research aimed at their identification in sweat.
- Table 4 - sensibility should be sensitivity - this is correct in Table 5.
Response: We thank the reviewer for this observation.
We have corrected the word.
- For references, there are a number of studies using urinary volatiles for CRC detection, these should also be quoted.
Response: We thank the reviewer for this observation. In the revised manuscript, we have incorporated additional studies in the Introduction that address urinary volatiles in CRC detection. Furthermore, we highlight that several investigations have reported the identification of biomarkers in different biological matrices, including urine, breath, and feces, for the diagnosis of CRC. References: Breath [13-28], urine [29-32], fecal [14],[33-34], [35], [36-37],[38-40].
Round 2
Reviewer 2 Report
Comments and Suggestions for Authors
The authors have made considerable changes to the manuscript and as a result it is much clearer. The methods and discussion have been improved.